# Large-Scale Expansion of Suspension Cells in an Automated Hollow-Fiber Perfusion Bioreactor

**DOI:** 10.3390/bioengineering12060644

**Published:** 2025-06-12

**Authors:** Eric Bräuchle, Maria Knaub, Laura Weigand, Elisabeth Ehrend, Patricia Manns, Antje Kremer, Hugo Fabre, Halvard Bonig

**Affiliations:** 1Institute for Transfusion Medicine and Immunohematology, German Red Cross Blood Service Baden-Württemberg-Hessen, 60528 Frankfurt, Germany; l.weigand@blutspende.de (L.W.); e.ehrend@blutspende.de (E.E.); p.manns@blutspende.de (P.M.); h.boenig@blutspende.de (H.B.); 2Faculty of Biological Science, Goethe University, 60439 Frankfurt, Germany; 3Terumo Blood and Cell Technologies, 1930 Zaventem, Belgium; maria.knaub@terumobct.com (M.K.); antje.kremer@terumobct.com (A.K.); hugo.fabre@terumobct.com (H.F.); 4Faculty of Medicine, Goethe University, 60528 Frankfurt, Germany; 5Department of Medicine, Division of Hematology and Oncology, University of Washington, Seattle, WA 98103, USA

**Keywords:** bioreactor, quantum, manufacturing, cell therapy, automation, cell expansion

## Abstract

Bioreactors enable scalable cell cultivation by providing controlled environments for temperature, oxygen, and nutrient regulation, maintaining viability and enhancing expansion efficiency. Automated systems improve reproducibility and minimize contamination risks, making them ideal for high-density cultures. While fed-batch bioreactors dominate biologics production, continuous systems like perfusion cultures offer superior resource efficiency and productivity. The Quantum hollow-fiber perfusion bioreactor supports cell expansion via semi-permeable capillary membranes and a closed modular design, allowing continuous media exchange while retaining key molecules. We developed a multiple-harvest protocol for suspension cells in the Quantum system, yielding 2.5 × 10^10^ MEL-745A cells within 29 days, with peak densities of 4 × 10^7^ cells/mL—a 15-fold increase over static cultures. Viability averaged 91.3%, with biweekly harvests yielding 3.1 × 10^9^ viable cells per harvest. Continuous media exchange required more basal media to maintain glucose and lactate levels but meaningfully less growth supplement than the 2D culture. Stable transgene expression suggested phenotypic stability. Automated processing reduced hands-on time by one-third, achieving target cell numbers 12 days earlier than 2D culture. Despite higher media use, total costs for the automated were lower compared to the manual process. Quantum enables high-density suspension cell expansion with cost advantages over conventional methods.

## 1. Introduction

Bioreactors are essential tools for large-scale cell cultivation, ranging from small bench-top systems to industrial-scale bioreactors exceeding 10,000 L. Regardless of size or mode of operation, their primary function is to create an environment that controls critical parameters such as temperature, osmolarity, pH, oxygen levels, and nutrient delivery—all crucial for maintaining cell viability and functionality [1]. Automated bioreactor systems have inherent advantages over manual processes, including improved reproducibility, minimized contamination risks, and reduced labor costs. They ensure continuous operation and allow seamless adaptation to non-diurnal schedules. By enabling precise control of culture conditions, possibly even with integration of feedback loops, these systems are particularly suited for achieving high cell densities and ensuring consistent product quality [2]. However, fine-tuning automated processes is challenging, especially in balancing nutrient supply and waste removal. Accumulated metabolic byproducts, such as lactate, can cause cellular stress, negatively affecting viability and functionality.

The selection of an appropriate bioreactor type has become increasingly critical for successful bioprocessing. Fed-batch bioreactors are widely regarded as the state-of-the-art for producing biologics like proteins and monoclonal antibodies. While fed-batch systems are advantageous in terms of simplicity, scalability, and control, they lack the sustained productivity and resource efficiency of continuous systems [3,4]. In fed-batch processes, harvesting typically involves the removal of the entire culture, including cells and spent media, making them single-use operations. This contrasts with continuous systems or perfusion cultures, which allow for extended production by retaining cells for subsequent growth cycles and multiple harvests [5].

Conventionally, cell culture in pharmaceutical manufacturing primarily focused on expanding producer cells for recombinant protein production in bioreactors, prioritizing yield over cell well-being. The advent of cell therapy, whereby the cells are the pharmaceutical product, has shifted the focus from secreted proteins to the cells themselves. This shift has given rise to new challenges, including the question of how similar is similar enough in terms of cell phenotyping. The manufacturing of many cell therapy medicines includes an ex vivo expansion phase for which bioreactors may be useful [6]. Mesenchymal stromal cells (MSCs), an adherent-growing cell type, exemplify this trend. These cells have been successfully expanded in both fed-batch cultures and in perfusion systems, including the hollow-fiber bioreactor of the Quantum Cell Expansion System. Quantum provided higher cell yields while maintaining phenotypic and functional similarity when compared with manually expanded MSCs, supporting the pharmacological similarity of cells generated by either process [7,8]. This system facilitates continuous media exchange to remove metabolic byproducts and supply critical nutrients, namely glucose. The Quantum hollow-fiber bioreactor’s semi-permeable capillary membranes create two physically separate compartments: the intracapillary (IC) space where the cells and large molecules reside, and the extracapillary (EC) space, for additional media flow and gas supply (Appendix A). Uniform mixing of cells and their surrounding media to improve nutrient distribution is promoted by adjustable media flow rates outside and inside the capillaries [9,10]. The semi-permeable fibers allow the exchange of smaller molecules, such as sugar and amino acids, gases, and waste products while retaining larger molecules, including most cytokines, and cells. Nankervis et al. investigated the fate of growth factors and cytokines and showed that compounds as small as 15 kDa, such as IL-2, were retained in the IC loop [11].

The adaptability of hollow-fiber perfusion systems, which allows for precise adjustments of flow directions and rates, suggests their potential utility in a wider range of applications. Furthermore, the closed, modular design of these bioreactors enhances safety and regulatory compliance, particularly for therapeutic cell production [12]. We hypothesized that non-adherent cell lines can also be efficiently expanded on an industrial scale using the Quantum Cell Expansion System, and here we present data supporting this hypothesis. We further propose that the system’s suitability could extend to the production of other suspension cell lines in allogeneic approaches, such as CAR-expressing NK92 cells, which require large doses (e.g., 1 × 10^8^ cells) and consistent quality across batches [13].

## 2. Materials and Methods

### 2.1. Conventional 2D-Cell Culture

Mouse erythroleukemia MEL-745A cl. DS19 (MEL) suspension cells (DSMZ, No. ACC 501), engineered to express human CD38, also known as *Darasorb* cells [14], allowing for phenotype analysis during cell expansion, were used for all pre-processing experiments. Cultures were maintained in DMEM high-glucose media (Gibco, Eggenstein-Leopoldshafen, Germany) supplemented with 20% fetal calf serum (FCS, Sigma-Aldrich, Darmstadt, Germany) and 1% penicillin-streptomycin (Gibco) at 37 °C in a humidified atmosphere with 5% CO_2_. Cells were passaged with a 1:10 ratio every 3–4 days at a density of 2–2.5 × 10^6^ cells/mL to ensure exponential growth. For expansion experiments, cells were seeded at 2 × 10^5^ cells/mL in T175 flasks (Sarstedt, Nümbrecht, Germany) four days prior to inoculation. At the conclusion of each culture, cells were harvested by centrifugation at 400× *g* for 5 min, washed in phosphate-buffered saline (PBS; Gibco), and counted, with viability determined using a hemocytometer or NucleoCounter (ChemoMetec, Allerod, Denmark).

Murine A0.01 T-cells (from Dr. H. T. He, Centre d’Immunologie de Luminy, Marseille, France) were cultured in RPMI-1640 media (Gibco) supplemented with 10% FCS and 1% penicillin-streptomycin at 37 °C in a humidified atmosphere with 5% CO_2_. Cells were subcultured every 3–4 days at a density of 1.5–2 × 10^6^ cells/mL to ensure exponential growth. Expansion and harvesting followed the same procedure as for MEL cells.

### 2.2. Manual Cell Expansion

On day 0, suspension cells (2.4 × 10^7^) were prepared in 120 mL of their respective culture media and seeded at a density of 2 × 10^5^ cells/mL into 4 × T175 flasks (Sarstedt), with each flask containing 30 mL of media (0.17 mL/cm^2^). The cells were incubated at 37 °C in a humidified atmosphere with 5% CO_2_ for three days. After incubation, all cells were collected, washed once with PBS (Gibco), and counted. For expansion, 33 × T175 flasks were seeded with 30 mL of culture media at 2 × 10^5^ cells/mL. Cells were harvested after four days or upon reaching a density of 2.5–3 × 10^6^ cells/mL. A total of 6.6 × 10^7^ cells were used to re-seed another set of 33 × T175 flasks for continued expansion. Any remaining cells were transferred to a sterile container for post-harvest processing (n = 1).

### 2.3. Automated Cell Expansion Using the Quantum Cell Expansion System

On day 0, suspension cells (5 × 10^7^) were prepared in an inlet bag containing 200 mL of culture media with 50% of the standard FCS concentration of the corresponding cell line (10% for MEL and 5% for A0.01). The cells were loaded into the Cell Expansion Set of a Quantum Bioreactor (hardware, software, and consumables from Terumo BCT, Lakewood, CO, USA), where they were continuously fed with FCS-free media from the EC side and with media containing 10% of the standard FCS concentration on the IC side. Cells were recirculated within the IC loop for four minutes every 24 h to ensure homogeneous distribution, and then repositioned into the bioreactor chamber with 300 mL of IC media. During the expansion phase, IC media was continuously supplied at an initial flow rate of 0.2 mL/min, which was increased to 0.4 mL/min 24 h prior to each harvest. The IC circulation rate was consistently maintained at negative half the inlet rate to facilitate the placement of the cells within the hollow fibers of the bioreactor during the expansion phase. The EC inlet flow rate was gradually increased from 0.0 to 0.2 mL/min 2–3 days prior to harvest, and further raised to 0.3 mL/min 24 h before harvest. At each harvest, a sample was taken from the IC loop during a recirculation step for cell enumeration and metabolite testing. On average, 83% of the cells were harvested to maintain a continuous culture with concentrations of 3 × 10^6^ to 5 × 10^6^ cells/mL, 4 or 3 days prior to the next scheduled harvest (7 harvests per run, n = 4), respectively. The final harvest was conducted using the automated harvest function of the system. Harvested cells were transferred into sterile containers for post-harvest processing.

### 2.4. Post-Harvest Processing

Harvested cells were counted and viability assessed, centrifuged at 400× *g* for 5 min, washed with PBS, and prepared for downstream applications, including flow cytometry and functional assays.

### 2.5. Metabolite Testing

A sample was drawn from the IC loop during recirculation for intermittent harvests or via the sample port on the EC side of the bioreactor unit when media consumption was analyzed daily in process development experiments. Following centrifugation at 13,000 rpm for one minute, glucose and lactate concentrations were measured using Contour XT (Bayer, Leverkusen, Germany) and Accutrend Plus (Roche, Penzberg, Germany), respectively.

### 2.6. Flow Cytometry

High transgenic surface expression of a human antigen (CD38) permitted quantitative assessment of cellular phenotype over time. Fluorescence-coupled clone HIT2 was used in combination with forward and side scatter on a FACSFortessa (BD Biosciences, San Jose, CA, USA) flow cytometer. Mean fluorescence intensity (MFI) of CD38-positive cells was assessed and normalized to the MFI of unconjugated cells.

### 2.7. Statistics

Descriptive statistics were calculated in Excel (Microsoft, Redmond, WA, USA), and graphics were generated in PowerPoint (Microsoft) and GraphPad Prism v. 9.5.1 (GraphPad, San Diego, CA, USA).

## 3. Results and Discussion

We developed and present here a protocol for the automated expansion of suspension cells using the Quantum Cell Expansion System, capable of producing a target dose of 2.5 × 10^10^ cells within a 29-day period (Figure 1). During preliminary experiments (n = 4), settings such as flow rates and circulation time were identified to ensure adequate nutrient supply to the cells. The process within the hollow-fiber perfusion bioreactor was modified from manufacturer-recommended settings to allow incremental harvests in a closed system, thereby maximizing total cell yield from a single cartridge per run.

Continuous ramped media exchange supported cell densities 15–20 times higher than those achievable in manual culture, averaging 3.1 × 10^7^ cells/mL on harvest days, with a mean viability of 91.3% (Figure 2A). The higher cell density in the bioreactor was demonstrated using the process-establishing MEL cell line (n = 3) and confirmed with an additional suspension cell line, A0.01 (n = 1). At concentrations beyond 3 × 10^7^/mL, nutrient consumption could no longer be compensated by increasing media flow rates without incurring prohibitively high media usage, rendering the process economically unfeasible. Confirmation of these basic process outcomes with two distinctly different suspension cell lines (MEL and A0.01) suggests general applicability of the established protocol for the expansion of rapidly proliferating cells. While other bioreactor systems can achieve similar or even higher cell concentrations [15,16], our process demonstrates satisfactory long-term cell viability for over 4 weeks of cultivation. One reason for this is that, compared to fed-batch and adherent cell cultures, our automated protocol enables multiple harvests with high reproducibility, offering a continuous and scalable production process rather than a single batch output [17].

Glucose is a critical nutrient for cell growth in culture. We expected continuous media exchange to maintain sufficiently high glucose concentrations inside the capillaries, while lactate is depleted through the circulation of the EC media as well as buffered by the basal media. To achieve this, various media flow rates were systematically evaluated during process development, with both glucose and lactate concentrations monitored daily; however, their levels remained predictable, so that measurements were subsequently limited to harvest days. Pilot experiments had identified a residual glucose concentration of approximately 1.5 g/L as the critical lower threshold for MEL cell growth. Adjusting the media flow rates to account for cell expansion kinetics, we established a protocol with maximum total flow rates of 0.4 mL/min IC media and 0.3 mL/min EC media 24 h before harvest (equivalent to 5–8 complete media exchanges per day) to maintain adequate glucose levels (above 1.5 g/L for MEL cells) and lactate levels below 26 mmol/L (Figure 2B). Our data thus essentially reproduce data reported by Lambrechts et al. with human periosteum-derived stem cells (hPDCs) expanded in a similar hollow-fiber bioreactor, where lactate concentrations did not return to baseline levels with similar flow rates to our study over an 8-day period [18]. Although media exchange rates could not fully restore glucose to initial levels or suppress lactate accumulation, Lambrechts et al. observed no adverse effects on viability or growth. These findings could be of interest for ex vivo CAR-T cell expansion, as due to metabolic stress at excessive cell density they lose effector functionality [19].

The cell expansion process was improved to allow biweekly harvests over four weeks, yielding an average of 3.1 × 10^9^ cells per harvest in approximately 180 mL (Figure 2C). This represents a 5-fold reduction in processing volume compared to manual expansion in culture flasks (Table 1), which translates into a shorter hands-on time in the processing of the product. Longer cultures beyond 4 weeks were not investigated.

In addition to cell growth and medium consumption, the performance of the expanded cells is a critical factor for the applicability of this technology. Satisfactory viability at high cell densities was demonstrated. Moreover, as an approximation of cell phenotype, at each harvest, we analyzed the expression density of a highly overexpressed (approx. 300,000 copies per cell) human surface antigen, namely CD38 [14]. As exemplified in Figure 2D, transgene expression remained stable throughout the entire four-week expansion period. Phenotypic similarity of harvested cells was likewise observed in two contemporary comparative studies of manual and automated expansion of adherent [20] and suspension cells [12].

Concurrently, cells were expanded manually. They reached average maximum concentrations of 2.8 × 10^6^ cells/mL. Viability remained at 98% and the phenotype was the same as in the bioreactor. Average doubling time was 25 h, as opposed to 20 h in the automated process, potentially due to differences in cell harvesting. Each passage omits a cell division, which may be less impactful in the automated system, as a fraction of cells remains undisturbed in the bioreactor. In contrast, manual harvesting requires the complete removal of all cells from the 2D cell culture and incubator environment.

In manual cultivation, cells received fresh media only during harvesting and culture continuation, while the bioreactor supplied cell culture with continuous media inflow. That is why, to generate 1 × 10^9^ cells with the manual protocol, we need 302 mL media and 75 mL FCS (Figure 2E). Since FCS is trapped in the inner circuit of the bioreactor where it is even concentrated over time, the outside circuit media does not need to be supplemented with FCS, so that while we need 3.5-fold more media, FCS consumption is reduced by 80% (1057 mL media, 15.5 mL FCS/1 × 10^9^ cells). For cells with requirements for specific growth factors, whether these will likewise only need to be added to the inner circuit depends on their molecular weight and will need to be confirmed experimentally [21]. These additives might need to be provided at similar concentrations in both manual and automated expansion systems [5]. The expansion of CAR-T cells in the hollow-fiber bioreactor, however, has already been shown to require cytokines exclusively in the IC media [12]. The need for consistent and appropriate supplementation across both methods ensures that cell growth and function are maintained, but it may also affect overall cost-efficiency in larger-scale applications.

To assess whether automation saves labor, the duration of each manual step in both processes was measured. Tasks were categorized as either common to both processes (e.g., post-harvest cell processing) or process-specific (e.g., manual vs. automated harvesting). On harvest days, hands-on time for the entire process was 2 h for automated, compared to 3 h for manual manufacturing (Figure 2F). Due to higher cell concentrations in the automated process, cell harvesting itself took five times longer for manual cell culture, i.e., 38 versus 7 min. The automated process achieved the same cell dose 12 days earlier than the manual method. Thus, the cumulative hands-on time was reduced by one-third, with the automated system requiring 21 h compared to 33 h for manual expansion. This confirms predicted efficiency gains in both time and labor associated with automated cell culture systems. The amount of labor that can be saved by transitioning from a manual to an automated system will vary, in part depending on the similarity of the manual and automated processes [17]. Hollow-fiber expansion being technically quite distinct from culture flasks, a considerable similarity validation effort will be expected if therapeutic cell processing is moved from one to the other.

Despite substantial acquisition and material costs, implementation of automated bioreactors has been shown previously to be a cost-efficient way to manufacture cell products [21]. In our hands, a comparison of consumable and media costs indicated that the automated process was more cost-effective than the manual approach. However, for our purpose, depreciation for incubators and hoods used in both processes or the Quantum Cell Expansion System (contributed by Terumo BCT) were not included in the calculation of cost of goods. As we are showing, savings for FCS and reduced labor costs more than offset the expense for the Quantum cartridge (Table 1). The functionally closed system also reduces contamination risks, enhancing both product safety and process reliability. Limitations of the work should be acknowledged: Our study was conducted using an immortalized, relatively low-maintenance suspension cell line. Further testing and improvement might be needed to achieve similar outcomes with more fastidious cell lines or primary cells. Future studies should aim to validate the system using primary human therapeutic cell lines in order to comprehensively assess its scalability, robustness, and clinical applicability beyond the murine cell line model used in this study.

## 4. Conclusions

By tailoring media compositions and flow rates, the Quantum Cell Expansion System facilitates the expansion of suspension cells to high concentrations while maintaining cell viability and phenotype. In this study, we reproducibly generated diagnostic cells, demonstrating the system’s efficiency and robustness in a controlled setting. The established process demonstrated clear advantages of automation over manual preparation, particularly in terms of increased safety through the reduction in open handling steps and decreased labor intensity. The use of a second, albeit also continuously growing, immortalized suspension cell line confirmed that many of the tested procedures are transferable, indicating that future expansions with immortalized or even primary human lines may require less establishment work. This supports the feasibility of generating large cell quantities with reduced effort and risk, enabling predictable, time-efficient bioprocess planning. Translating these findings to cell therapy medicines will require further validation, particularly regarding scalability, consistency, and regulatory compliance. With these caveats, we propose that Quantum holds promise for the suspension cell expansion of therapeutic cells.

## Figures and Tables

**Figure 1 bioengineering-12-00644-f001:**
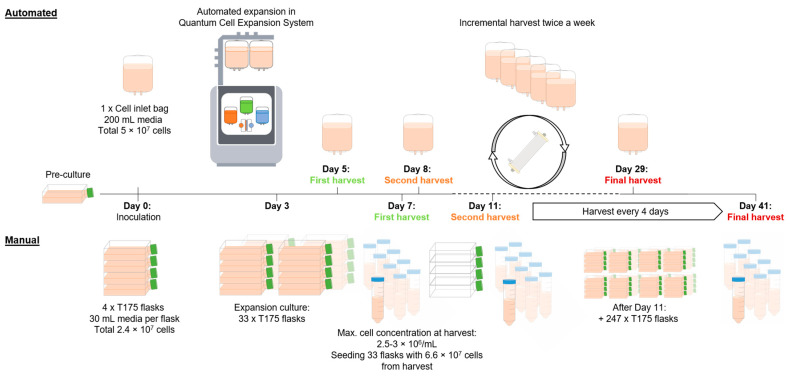
A schematic overview of the automated and manual production of 2.5 × 10^10^ suspension cells. This schematic illustrates key procedural differences between manual and automated cell expansion workflows. The automated process (top) leverages the Quantum Cell Expansion System. It begins with the inoculation of 5 × 10^7^ cells, followed by continuous automated expansion with intermittent harvests every 3 to 4 days. The automated process reaches the targeted cumulative cell dose of 2.5 × 10^10^ with the final harvest on day 29. In the manual process (bottom), cell expansion is initiated by inoculating 4 × T175 flasks with a total of 2.4 × 10^7^ cells. On day 3, the accumulated cells are transferred into a total of 33 × T175 flasks, with a defined cell density of 6 × 10^6^ cells in 30 mL media per flask. This is followed by harvesting the cells and inoculating 33 new flasks for continuous expansion, a step that is repeated every 4 days, reaching the targeted dose of 2.5 × 10^10^ with the harvest on day 41.

**Figure 2 bioengineering-12-00644-f002:**
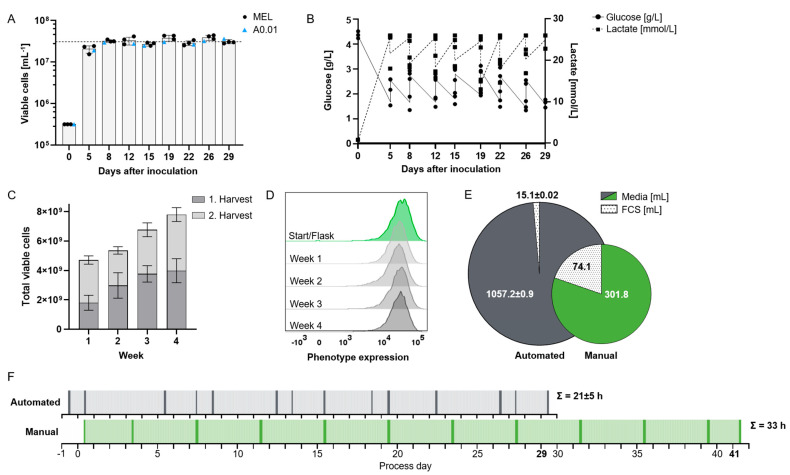
Establishment of automated feeding protocol for high-yield production of suspension cell lines. (**A**) Viable cell concentration at harvest for two suspension cell lines, MEL (n = 3; used for establishing the expansion protocol) and A0.01 (n = 1). Mean of peak concentration 31 × 10^6^ cells/mL (dotted line). (**B**) Glucose and lactate concentrations at harvest and after re-suspension of residual post-harvest MEL cells (n = 3). (**C**) Cumulative viable cell yield over four weeks, with two harvests per week of both cell lines (n = 4). (**D**) Flow cytometric assessment of transgenic expression of a surface molecule as representative marker of phenotypic stability during expansion in bioreactor for MEL cells. (**E**) Consumption of media and fetal calf serum (FCS) normalized to production of 1 × 10^9^ cells comparing automated (n = 3) and manual process. (**F**) Timeline for producing 2.5 × 10^10^ suspension cells using automated (top) (n = 3) vs. manual (bottom) processes. Labor is shown in 30 min increments: hands-on time (dark), unobserved work (light). Cumulative hands-on time is indicated for each process.

**Table 1 bioengineering-12-00644-t001:** A selected costs calculation, comparing automated and manual processes for the production of 2.5 × 10^10^ MEL suspension cells. Data for the automated process were averaged from three independent runs, and those for the manual process were extrapolated from reduced-scale runs.

	Manual	Automated
**Media to generate 2.5 × 10^10^ cells [mL]**		
Basal media	7757	27,170
FCS	1905	388
**Open steps**	1636	1
**Volume processed per harvest [mL]**	990	177
**Process unique labor [h]**	19.4	6.9
**Cost [EUR]**		
Consumables	5385	5021
Labor (EUR 60/h)	2088	1206
Total	7473	6227
For 1 × 10^9^ cells	298	242

## Data Availability

The original contributions presented in this study are included in the article/Appendix A. Further inquiries can be directed to the corresponding author.

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
