# Peer review of "Large-Scale Expansion of Suspension Cells in an Automated Hollow-Fiber Perfusion Bioreactor"

_bioengineering, 2025, doi:10.3390/bioengineering12060644_

Round 1

Reviewer 1 Report

Comments and Suggestions for Authors

I am delighted to review the manuscript entitled „Large-scale expansion of suspension cells in an automated hollow-fiber perfusion bioreactor“.

Congrats to the authors for the very well-written manuscript. The text is well-structured, with a strong focus and scientific impact on bioengineering in cell therapy.

I have just a few minor questions:

-Please provide more informations for the here applied hardware (provide a technical depiction, scheme etc.).

-Do the authors have experience with human cell lines in terms of the here described experimental set-up? Please comment!

Reviewer 2 Report

Comments and Suggestions for Authors

please see the attached file.

Reviewer 3 Report

Comments and Suggestions for Authors

This research article effectively describes the cultivation of suspension cells using an automated hollow-fiber perfusion bioreactor. It is commendable that the experiments are well-designed, and the results align with the experimental design, providing interesting information. While most of the manuscript's content is sound, a few points could be further improved to enhance its overall completeness and impact. These are:

  1. Abstract: It is recommended that the abstract be merged into a single paragraph.
  2. Materials and Methods: To enhance the credibility and clarity of the methodology, the number of experimental replications for each experiment should be clearly stated. Specifying the number of replications is crucial for assessing the precision and reproducibility of the results.
  3. Conclusion: The conclusion would be strengthened and offer greater value if it were adjusted to explicitly highlight the key findings of this study. Furthermore, it should clearly indicate how these findings can be applied to future research or practical applications.
